# Views of healthcare professionals regarding barriers and facilitators for a Fracture Liaison Service in Malaysia

Min Hui Cheah[1]☯, Pauline Siew Mei Lai[2,3]☯, Terence Ong🄳[1]*

1 Department of Medicine, Faculty of Medicine, University of Malaya, Kuala Lumpur, Malaysia, 2 Department of Primary Care Medicine, Faculty of Medicine, University of Malaya, Kuala Lumpur, Malaysia, 3 School of Medical and Life Sciences, Sunway University, Subang Jaya, Malaysia

☯ These authors contributed equally to this work.
* terenceong@doctors.org.uk

**Data Availability Statement:** All relevant data are within the manuscript.

**Funding:** This work was supported by research funds provided by the Amgen Biopharmaceuticals Malaysia Sdn Bhd (DONATION-329489). The

## Abstract

This study aimed to explore the views of healthcare professionals regarding the barriers and facilitators for a Fracture Liaison Service (FLS) in Malaysia. The qualitative study was conducted from February to December 2021 at a tertiary hospital in Malaysia. Doctors, nurses, pharmacists, and policymakers were recruited via purposive sampling. Semi-structured in-depth interviews were conducted until thematic saturation was achieved. Data were transcribed verbatim and analysed using thematic analysis. Thirty participants [doctors (n = 13), nurses (n = 8), pharmacists (n = 8), and policymakers (n = 1)] with 2–28 years of working experience were recruited. Three themes emerged: 1) Current delivery of secondary fracture prevention; 2) Importance of secondary fracture prevention, and 3) FLS sustainability. Some participants reported that the current post-hip fracture care was adequate, whilst some expressed concerns about the lack of coordination and continuity of care, especially in non-hip fragility fracture care. Most participants recognised the importance of secondary fracture prevention as fracture begets fracture, highlighting the need for a FLS to address this care gap. However, some were concerned about competing priorities. To ensure the sustainability of a FLS, cost-effectiveness data, support from relevant stakeholders, increased FLS awareness among patients and healthcare professionals, and a FLS coordinator were required. Training and financial incentives may help address the issue of low confidence and encourage the nurses to take on the FLS coordinator role. Overall, all participants believed that there was a need for a FLS to improve the delivery of secondary fracture prevention. Addressing concerns such as lack of confidence among nurses and lack of awareness can help improve FLS sustainability.

## Introduction

Fragility fractures are fractures resulting from low-energy trauma such as falls from standing height or less [1]. Patients who sustained fragility fractures often have underlying osteoporosis

funders had no role in study design, data collection and analysis, decision to publish, or preparation of the manuscript.

**Competing interests:** The authors have declared that no competing interests exist.

**Abbreviations:** FLS, Fracture Liaison Service; HCP, Healthcare Professional; TPB, Theory of Planned Behaviour.

and are at risk of subsequent falls or fractures [2]. While fragility fractures are associated with significant morbidity and mortality, there remains a gap in the delivery of secondary fracture prevention [3, 4]. A Fracture Liaison Service (FLS) is a coordinator-based model developed to reduce subsequent fractures and mortality by increasing the diagnosis of osteoporosis and improving the treatment of osteoporosis [5, 6]. FLS was first introduced in Glasgow, Scotland in 1999 [6, 7]. Generally, FLS involves proactive identification of fragility fracture patients, bone health investigations, osteoporosis treatment initiation where necessary, and follow-up to improve treatment adherence [6].

Despite the known benefits of a FLS, its uptake and implementation have not been universal. A review of the secondary fracture prevention service in Singapore highlighted insufficient resources and a lack of knowledge on the importance of FLS as a barrier to its implementation [8]. Secondary fracture prevention is not a top priority for policymakers; hence, osteoporosis is not included in Singapore's chronic disease management program [8].

Several qualitative studies regarding the implementation of a FLS have been conducted in Canada and the United Kingdom [9–12]. Some were done post-implementation [9, 11, 12], whilst one was conducted both at baseline and post-implementation to gauge how views changed over time [10]. Most studies addressed challenges and facilitators of a successful FLS implementation [10–12], whilst one study reported on strategies needed to develop a business case for the implementation of a FLS [9]. Interviews with healthcare professionals (HCPs) and members of the FLS implementation steering committee, such as managers and social workers, revealed that fracture prevention coordinators were the key to successful implementation as they helped in coordinating and promoting multidisciplinary teamwork [11, 12]. In addition, demonstrating the need for a FLS, providing resources and selecting an appropriate site for a FLS were reported as facilitators for FLS implementation [10].

In Malaysia, the delivery of secondary fracture prevention care has been reported to be lacking [4, 13]. Lack of clarity regarding the team responsible for managing osteoporosis was one of the osteoporosis treatment care gaps [14]. FLS can help address this problem as it improves coordination between departments [6]. Although a previous qualitative study was done to identify strategies for implementing a FLS [10], findings from this study may not be applicable as their healthcare system differs from Malaysia. To date, there is a paucity of data regarding the views of relevant stakeholders on the current delivery of secondary fracture prevention and their willingness to participate in or implement a FLS from a middle-income developing country in Asia. Therefore, this study aimed to explore the views of healthcare professionals regarding the barriers and facilitators for a FLS in Malaysia, especially to facilitate its implementation in this tertiary hospital.

## Materials and methods

### Study design

This qualitative study was conducted at a tertiary hospital in Malaysia from 1$^{st}$ February to 31$^{st}$ December 2021.

### Theory

The theory of planned behaviour (TPB) was selected as the conceptual framework in this study as it could help predict stakeholders' intention to set up a FLS and understand factors that either encourage or hinder them from implementing a FLS [15] (Fig 1). The theory proposes that an individual's intention and behaviour (the setting up of a FLS) are driven by three major constructs: attitude, subjective norms, and perceived behavioural control. Attitude refers to an individual's positive or negative evaluation of performing a behaviour. In our study, we

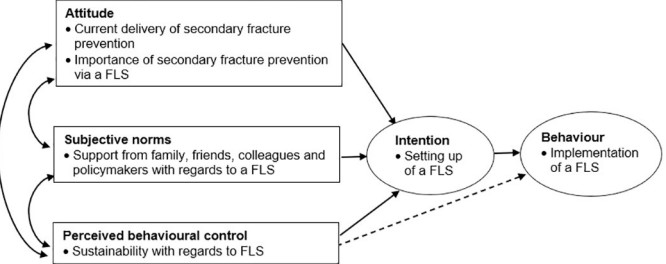

**Fig 1. Conceptual framework based on the Theory of Planned Behaviour.**

assessed stakeholders' attitudes by exploring their opinions on the current delivery of secondary fracture prevention and the perceived importance of implementing an FLS. Subjective norm involves the perceived social pressure to perform or not perform a behaviour. We investigated this construct by understanding stakeholders' perceptions of whether significant others (such as family, friends, and colleagues) support or oppose the implementation of the FLS model. Perceived behavioural control is the perceived ease or difficulty of performing a behaviour, reflecting past experiences and anticipated obstacles. We identified this by examining the barriers and facilitators stakeholders perceive in implementing and sustaining the FLS model. These constructs together shape stakeholders' intentions, indicating their willingness and readiness to implement and sustain the FLS model.

## Participants

Four groups of participants (doctors, nurses, pharmacists, and policymakers) who could speak either English or Malay were recruited. Doctors were included as they were primarily responsible for the management of patients with fragility fractures, while nurses were recruited because of their direct involvement in caring for these individuals and, in several centres, their role as FLS coordinators. Pharmacists were included as potential candidates for the FLS coordinator position. Given that our study aimed to explore the perceptions regarding a new service establishment, policymakers who could influence practice were also recruited.

**Doctors.** Geriatricians, endocrinologists, family medicine physicians, and orthopaedic surgeons who were involved in the management of patients with fragility fractures were recruited. Doctors from different clinical backgrounds were recruited as they were involved in various aspects of fragility fracture patient management (e.g. acute management, long-term follow-up, primary care, and secondary care) and may provide different perspectives. Specialists with <1 year of experience in treating these patients were excluded.

**Nurses.** Nurses working in orthopaedic wards, who were directly involved in the care of patients with fragility fractures were recruited. Nurses with <1 year of experience in treating these patients were excluded.

**Pharmacists.** Pharmacists who were involved in the dispensing and counselling of medications for osteoporosis were recruited. Trainee pharmacists were excluded.

**Policymakers.** A policymaker is defined as a person involved in formulating policies, and who has the authority to influence and change practice. In our setting, policymakers were the Director and the Deputy Directors of the hospital, and the relevant Head of Departments involved in the care of patients with fragility fractures.

**Table 1. Topic guide for semi-structured in-depth interviews.**

| |
|---|
| 1. How is the delivery of secondary fracture prevention currently? |
| 2. Have you ever heard of the term Fracture Liaison Service? |
| 3. What do you think should be the functions of a Fracture Liaison Service? |
| 4. Is there a need for a Fracture Liaison Service? |
| 5. What are the difficulties that might be faced when implementing the Fracture Liaison Service? |
| 6. Who should be involved in the Fracture Liaison Service? |
| 7. Is there a need for a Fracture Liaison Service coordinator? |
| 8. Which healthcare professional can serve as a Fracture Liaison Service coordinator? |
| 9. What factors do you think would facilitate the establishment of a Fracture Liaison Service in Malaysia? |

## Instruments used

Participants' basic information was collected using a baseline demographic questionnaire. A topic guide (Table 1) was used to guide interviews. The same topic guide was modified to cater for the different stakeholders (S1 Appendix). For policymakers, questions that might influence their decision-making process regarding a new service implementation were included. If the participants had never heard of the term "FLS" or secondary fracture prevention, it would be explained to them during the interviews.

## Recruitment

**Doctors.** Doctors were recruited through email or WhatsApp. Invitation notes were sent to the doctors' email addresses obtained from the hospital's website. At the same time, an invitation was disseminated through the personal contacts of the research team members. We purposively recruited doctors from different clinical expertise and with different years of experience to ensure maximum variation.

**Nurses.** Nurses were initially recruited through email. An email was sent to the nurse manager of the orthopaedic wards and disseminated to all nurses. However, participation was low. To encourage participation, the researcher approached the nurses directly at the orthopaedic wards instead. Nurses with varying years of working experience were recruited to ensure diverse perspectives.

**Pharmacists.** An invitation was sent through email and WhatsApp via the drug information pharmacist (our point of contact) to other pharmacists. Pharmacists from outpatient, inpatient, and drug information centre were purposively sampled for maximum variation.

**Policymakers.** The list of policymakers was selected based on the hospital's organisation chart. Policymakers were approached via email addresses obtained from the hospital's website. A second round of invitations was sent six weeks later to encourage participation.

## Procedure

For those who agreed to participate, an appointment to conduct the interview was set up. Informed consent was obtained, and baseline demographic details were collected. Semi-structured in-depth interviews were conducted virtually as this study was conducted during the COVID-19 pandemic. The interviews were conducted one-on-one with the participants by MHC, a postgraduate researcher who has undergone training in doing qualitative research. No relationship was established between the interviewer and the participants before the study commencement. The purpose of the interview was explained to the participants at the beginning of the interview. Interviews lasted 30 to 60 minutes. Field notes on non-verbal cues and

interview dynamics were recorded. All interviews were audio-recorded, transcribed verbatim, and analysed using thematic analysis. Analysis was conducted concurrently with ongoing interviews until thematic saturation was achieved.

### Data analysis

Participants' demographic details were summarised to maintain anonymity. Qualitative data were managed using Atlas.ti software version 22 (Scientific Software Development GmbH, Berlin, Germany). All audio recordings were transcribed by a transcriptionist, and cross-checked by MHC for accuracy and completeness. Interviewee transcript review was not performed. Researchers involved in the analysis were a geriatrician (TO), an academic pharmacist (PSML) and a postgraduate researcher (MHC). All researchers were conscious of their personal views and biases regarding FLS implementation. The team had regular meetings to reflect on and discuss the analysis.

MHC independently reviewed and coded the transcripts. A meeting was carried out after the first three transcripts were coded. PSML and TO reviewed the quotes and codes identified to ensure the reliability and consistency of the codes. The emergence of patterns from open coding was observed. MHC then continued to analyse the remaining transcripts. The research team met fortnightly to review the codes. The codes were merged into categories (subthemes) and a network was built to identify the relationships between codes, categories, and concepts derived from open coding. The subthemes were mapped based on the TPB as we identified a close relationship between the themes that emerged and the TPB. A draft codebook was generated and discussed among the researchers, and the subthemes and themes were further refined based on the discussions until the codebook was finalised. Disagreements were resolved through consensus.

### Ethical approval

Ethics approval was obtained from the University Malaya Medical Centre Medical Research Ethics Committee before the study (ID No: 202118–9684). All procedures performed in studies involving human participants were in accordance with the ethical standards of the institutional and/or national research committee and with the 1964 Helsinki Declaration and its later amendments or comparable ethical standards. Written informed consents were obtained from all individual participants included in the study.

## Results

A total of 13/20 doctors, 8/10 nurses, 8/10 pharmacists and 1/11 policymakers agreed to participate (response rate = 58.8%) (Table 2).

Three themes emerged from our data (Table 3). Themes and subthemes are presented in detail below (S2 Appendix).

### Current delivery of secondary fracture prevention

Some doctors, nurses and pharmacists thought that the current delivery of secondary fracture prevention was adequate as specialised services such as the orthogeriatric service [a medical-surgical model specifically designed for managing hip fracture patients, bringing together multidisciplinary health specialists from trauma, orthopaedics, and geriatric medicine [16]] was set up. However, a geriatrician expressed concern that patients with non-hip fractures may be neglected in the current system. Nurses and pharmacists reported that many patients who were admitted for hip fractures were started on medication for osteoporosis and falls

**Table 2. Baseline demographic characteristics of study participants.**

| Characteristics | Results |
|---|---|
| **Doctors (n = 13)** | |
| Mean age (range) | 38 (35–49) |
| Gender, n (%) | |
| Male | 4 (30.8) |
| Female | 9 (69.2) |
| Designation, n (%) | |
| Specialist | 10 (76.9) |
| Consultant | 3 (23.1) |
| Clinical Area of Expertise, n (%) | |
| Primary care | 4 (30.8) |
| Geriatric medicine | 4 (30.8) |
| Orthopaedics | 3 (23.1) |
| Endocrinology | 2 (15.4) |
| Length of working experience (range) | 12 (7–23) |
| **Nurses (n = 8)** | |
| Mean age (range) | 33.5 (25–48) |
| Gender, n (%) | |
| Male | 1 (12.5) |
| Female | 7 (87.5) |
| Length of working experience (range) | 12.3 (4–25) |
| **Pharmacists (n = 8)** | |
| Mean age (range) | 33.6 (27–43) |
| Gender, n (%) | |
| Male | 1 (12.5) |
| Female | 7 (87.5) |
| Unit, n (%) | |
| Outpatient | 3 (37.5) |
| Inpatient | 1 (12.5) |
| Drug information centre | 4 (50.0) |
| Length of working experience (range) | 8.8 (2–16) |
| **Policymaker (n = 1)** | |
| Mean age | 56 |
| Gender, n (%) | |
| Male | 1 (100) |
| Length of working experience | 28 |

prevention. The policymaker also believed that the current secondary fracture prevention delivery was of a high standard as patients were referred when necessary.

> "I will say (the current delivery of secondary fracture prevention) is very good . . . we have an ortho-geriatric service in the hospital which reviews all hip fractures cases . . . but I am not too sure whether other fractures are captured or not."

[*DR7, 38 y.o., Geriatrician*]

**Table 3. Themes emerged from qualitative data analysis.**

| Themes | Subthemes |
|---|---|
| Current delivery of secondary fracture prevention | - |
| Importance of secondary fracture prevention via a FLS | - |
| FLS sustainability | Cost-effectiveness |
|  | Support from relevant stakeholders needed |
|  | Awareness regarding FLS |
|  | Need for a FLS coordinator |
|  | Need for national policy |

Abbreviation: FLS—Fracture Liaison Service

*"If a patient has a femur fracture, we will give him some advice on the importance (of fall prevention) prior to discharge, we advise him to take care, . . . for someone to accompany them to the toilet . . . wear non-slip shoes. . ."*

[*NS2, 32 y.o., Nurse*]

*". . .I see a lot of physicians prescribing rocaltrol (calcitriol), calcium supplements . . . (and) bisphosphonates for osteoporosis. So I think we are actually doing good in . . . managing patients with osteoporosis."*

[*PS1, 36 y.o., Pharmacist*]

*"I can say that our fracture management in emergency medicine and by our orthopaedics people is quite efficient. I think they will refer to the appropriate primary team."*

[*PM1, 56 y.o., Policymaker*]

However, some participants also reported a care gap, as there was a lack of coordination between departments, leading to a lack of continuity of care.

*". . . it's (the delivery of secondary fracture prevention) still not enough . . . there are some patients who came in with a broken left leg . . . the next time they come in with their right leg (fractured) due to a fall."*

[*NS4, 34 y.o., Nurse*]

*"At the moment everybody is dancing their own orchestra, once you have a fracture, you will be under the orthopaedics team . . . it is up to the orthopaedics team to refer to the rehab team . . . or endocrine . . . having like a few clinics run different kind of show, you going to have many, many problems . . . The patient gets 'rojak (mixed advice)'."*

[*PS1, 36 y.o., Pharmacist*]

## Importance of secondary fracture prevention via a Fracture Liaison Service

All participants thought that with the implementation of a FLS, patients would receive better care to prevent future fractures as fracture begets fracture. The consequences of the subsequent fractures were often worse, causing direct and indirect burdens to patients and their families.

Participants believed that implementing a FLS could lead to a reduction in admissions related to subsequent fractures. However, the policymaker had some reservations regarding implementing a FLS as there was a need to balance the hospital's priorities at this current moment.

> *"It is definitely important because it is a known fact that there is a high risk of an elderly falling again . . . the more they get admitted, the more they get bedridden, the more surgeries they go through, they tend to deteriorate further, and the one-year mortality is high without surgery, and we found that even with surgery also, it's not like getting it down to zero."*

> [*DR12, 45 y.o., Orthopaedic surgeon*]

> *"The need is there. It's just that we need to balance on whether it is urgent or we can wait . . . But based on the current situation, we can wait for a while. But it's a good thing to have this because it will help the patients in the long term, it would help to reduce the cost of treating a patient."*

> [*PM1, 56 y.o., Policymaker*]

FLS could enhance coordination between departments, and improve HCPs and patients' education. These improvements could lead to enhanced quality of life for patients, as well as reductions in re-fracture rates and mortality. It can also improve the hospital's reputation.

> *". . . it (FLS) will be a one-stop centre for patients to seek information . . . for the doctor to treat the [fragility fracture] patients . . . a good way of centralising the service of . . . risk assessment, fall prevention and fracture prevention."*

> [*PS5, 39 y.o., Pharmacist*]

> *". . . educate the staff through FLS, then the staff will educate the patients."*

> [*NS5, 25 y.o., Nurse*]

> *". . . [FLS] many benefits. . . improve the wellbeing . . . potentially reduce healthcare cost . . . reduce the need to utilise acute inpatient services . . . serve as a form of awareness or education . . . potentially improve the quality of life of the patient."*

> [*DR4, 36 y.o., Geriatrician*]

> *"Being a teaching hospital, you must have a special kind of service. It also would boost the image of our hospital."*

> [*PM1, 56 y.o., Policymaker*]

### Fracture Liaison Service Sustainability

**Cost-effectiveness.**   Participants expressed the importance of demonstrating the cost-effectiveness of the FLS to convince the hospital's management of setting up a new service. In addition, FLS can lead to a registry functioning as a database for fragility fracture patients. A proper framework or workflow should be in place to ensure sustainability.

> *". . . cost-effectiveness is important. If you can run the service at a minimal cost, without causing a spike in the current usage of the unit, and that provides benefits . . . many people will*

*tend to follow. But if it is a very costly thing to do, no matter how effective it is, the moment the stakeholder has financial constrain, he will definitely cut down this service."*

[*DR3, 37 y.o., Geriatrician*]

*"... not sure how much cost is needed, but ... if you reduce admissions, you reduce the... cost of treatment, sure that is more we can save compared to what we have to put in the cost for this service."*

[*PS6, 27 y.o., Pharmacist*]

*"It has to have a database of all the patients with a fracture risk ... part of the function of FLS is to set up the framework for the FLS."*

[*PS5, 39 y.o., Pharmacist*]

*"I think the most important thing is to convince the policymaker ... with the evidence ... the fracture reduction is reduced... how much we can save the money."*

[*PM1, 56 y.o., Policymaker*]

**Support from relevant stakeholders needed.** Support from relevant stakeholders is the fundamental component to ensure the success of a service. Suggestions were made to include a multidisciplinary team comprising doctors, nurses, pharmacists, physiotherapists, occupational therapists, social workers, and administrative staff in the FLS team. Successful FLS stories can serve as powerful motivators and promotional tools for the service. There is a need to incorporate FLS into routine clinical settings.

*"A multidisciplinary team is needed ... (like the) physiotherapist or occupational therapist ... social workers to seek financing for patients ... nurses to give appropriate nursing plans ... Pharmacists to review medications of patients, and last but not least, the clinicians."*

[*DR7, 38 y.o., Geriatrician*]

*"Getting them (FLS stakeholders) to actually put it into writing, their experiences and their good feedback, I think this would be a good way to promote the service, not just in the towns but in rural areas that can access Fracture Liaison Service."*

[*DR1, 38 y.o., Primary Care Physician*]

One participant expressed the opinion that the FLS should not rely solely on an individual but rather should be integrated into routine clinical services.

*"... the state of our fracture liaison service here is still very much a champion-driven process ... we are not there yet, we have not established it such that it does not depend on an individual champion or individual physician."*

[*DR4, 36 y.o., Geriatrician*]

**Awareness regarding Fracture Liaison Service.** Many doctors were aware of FLS, but most pharmacists and nurses had never heard of FLS. An orthopaedic surgeon mentioned that a lack of awareness regarding the importance of secondary fracture prevention could potentially hinder FLS implementation. Some healthcare professionals placed less priority on

secondary fracture prevention compared to other management priorities and orthopaedic doctors were more focused on the surgical procedural part of the job. A change in mindset was required to deal with the increased fragility fracture burden.

*". . . it is the mindset of the (orthopaedic) doctors . . . They are interested in fixing a fracture, not doing much to prevent future fractures, so changing the mindset of a person is the one I find more difficult. We need to start educating the doctors, who are not aware of the importance of this tsunami of fractures, its importance is not given as great as the other fields of orthopaedics."*

*[DR13, 49 y.o., Orthopaedic surgeon]*

**Need for a Fracture Liaison Service coordinator.** All participants agreed that there was a need for a FLS coordinator to ensure the sustainability of the service. A FLS coordinator is needed to engage relevant stakeholders and to keep the service ongoing.

*"Most definitely (need for FLS coordinators) because ownership of that service needs to come from someone constant. I believe the coordinator is the most constant . . . at least two that can help to alternate their duties."*

*[DR1, 38 y.o., Primary care physician]*

Ideally, a new post should be created for a FLS coordinator. However, due to constraints in a resource-limited setting, the role will need to be delivered utilising current personnel.

*". . . if we have a specialist nurse, or . . . a full-time registrar . . . who could be trained for this purpose, we could create a post for this person . . . unfortunately, we are in a sub-ideal situation . . . getting extra staff is definitely very challenging at the current time. . . So, rearranging personnel. . . and prioritising will be the way to go."*

*[DR3, 37 y.o., Geriatrician]*

Participants held mixed opinions regarding who should be the FLS coordinator. Some participants thought that any HCP could become a FLS coordinator, as long as he/she had good communication skills and dedication. However, some doctors thought that nurses were more suited for the role, as they were well-versed with the hospital's system. A few nurses mentioned that they were not confident in becoming a FLS coordinator as they were used to following instructions rather than making decisions. Most nurses felt that doctors could perform the job better.

*". . . (hospital) nurses know how to obtain an appointment. . . and cost. . . They know how the electronic medical records work . . . nurses can do patient education. I think overseas, nurses play the main role in terms of patient education and counselling. But we just have to empower the nurses, train the nurses."*

*[DR9, 37 y.o., Primary Care Physician]*

*"I think it's possible (for a nurse to be a FLS coordinator), but we are used to receiving (instructions from doctors), we are not used to making our own decisions. If I become the coordinator, maybe others may not be confident (with my suggestions)."*

*[NS3, 34 y.o., Nurse]*

On the other hand, some nurses claimed that they were confident to become a FLS coordinator if given training and support.

*"Yes (becoming a FLS coordinator) . . . with a briefing of what he/she (the nurse) needs to do before starting the job . . . maybe with (working) experience . . . at least 3 to 5 years . . ."*

*[NS6, 48 y.o., Nurse]*

Pharmacists claimed they could fulfill the role of FLS coordinator but cited current understaffing as a challenge.

*"I don't have any problems with (pharmacists as FLS coordinator), provided that there is enough manpower."*

*[PS8, 30 y.o., Pharmacist]*

Incentives in terms of certification or promotion would help to encourage HCPs to work as a FLS coordinator.

*"the nurse will ask the prospects of FLS training, and why he/she should do more if there is no salary increment . . . and if you can provide accreditation (as a FLS nurse), and you provide a career pathway . . . there should be some incentives."*

*[DR5, 38 y.o., Endocrinologist]*

**Need for national policy.** Having a local FLS guideline on how to deliver the service and appropriate benchmarks could help support its implementation. Recognition and validation of the service at either a national or international level would help ensure the sustainability of the service.

*"Yes, Clinical Practice Guideline (CPG) is very important. I think CPG didn't mention much about Fracture Liaison Services, probably can improvise on that, add on this Fracture Liaison Services, and involve the multi-disciplinary approach."*

*[DR2, 38 y.o., Primary care physician]*

A function of FLS is to follow up with patients and refer them for long-term management of osteoporosis. However, at present, doctors at government health clinics in Malaysia cannot prescribe antiosteoporosis medications. This restriction prevents patients from being referred from hospitals to health clinics. Hence, a change in this policy is required to facilitate continuity of care for fragility fracture patients beyond the acute hospital.

*"They (primary care physicians at government health clinics) cannot prescribe any bone treatment at all . . . as much as I want to involve them, there need to be some changes in this first . . . they are ideal for the continuation of care, but I am hoping that this (change in prescribing rights) is done over the subsequent few years."*

*[DR4, 36 y.o., Geriatrician]*

## Discussion

This study contributes to the existing body of work by identifying the best way to implement FLS in a middle-income developing country in Asia by exploring the local stakeholders' perceptions of FLS implementation. The use of qualitative methods and the TPB was suitable as it helped identify the various perceptions of the stakeholders regarding FLS implementation and their willingness to participate in FLS.

Some participants deemed that the delivery of secondary fracture prevention was adequate for hip fracture patients, as they were co-managed by members of the orthopaedics and geriatric medicine team. However, patients with non-hip fragility fractures were not included, potentially resulting in a disparity in care. Despite a higher proportion of hip fracture patients being initiated on antiosteoporosis medications compared to wrist fracture patients, previous studies have shown a secondary fracture prevention gap in Malaysia [4, 13, 17, 18]. Our participants were unaware of this gap, especially the gap among patients with non-hip fractures. This lack of awareness could be a barrier to FLS implementation as participants did not perceive an urgent need for it. FLS was needed to help close the gap as it aims to cover all patients with fragility fractures.

The policymaker expressed that while FLS was important, it was not considered urgent. Our findings were similar to Singapore, where secondary fracture care prevention is also not a priority with policymakers, and therefore not included in their national policy [8]. In contrast, Canada's FLS is part of their health services network and receives good support as the fragility fracture population had been made a priority [10]. The lack of focus on FLS in our country may impose challenges in obtaining resources to run FLS.

Our participants thought that FLS had many benefits, and therefore supported the need for a FLS. Our findings concurred with previous studies that reported that FLS could improve treatment initiation and bone health investigations, reduce refracture rates and mortality [5]. The perceived importance and benefits of FLS would encourage the participants to set up and implement a FLS [15]. Communicating the benefits of FLS to patients and providers would facilitate its implementation [10].

FLS was perceived as cost-effective by our participants. Previous studies have also proven the cost-effectiveness of FLS regardless of the site of fractures [19–21]. Given the long-term reduction of re-fracture rates by FLS, the initial cost of implementation was considered negligible. Demonstrating cost-effectiveness data could help support FLS implementation in Malaysia.

Additionally, our participants emphasised the need for support from relevant stakeholders to ensure successful FLS implementation. A multidisciplinary team including doctors, nurses, pharmacists, dietitians, physiotherapists, occupational therapists, social workers and administrative staff should be involved, as supported by findings from other studies [10, 11]. A study in Alberta, Canada reported that establishing a provincial steering committee to facilitate learning and monitor FLS implementation was effective, providing a platform for experience sharing, discussion and decision-making [10]. In addition, regular multidisciplinary team meetings helped ensure service sustainability. Successful FLS stories from other facilities could further facilitate implementation. However, FLS should be tailored to the local context, and the involvement of local champions is essential [10, 12]. Nevertheless, our participants felt that the FLS should not be too champion-driven, but instead should be incorporated into routine clinical service.

Based on a previous study, orthopaedic surgeons were identified as key implementation stakeholders, given their primary role in managing fragility fracture patients. They could positively influence the perception of the orthopaedic team and other relevant stakeholders [12].

However, an orthopaedic surgeon mentioned that competing priorities among orthopaedic doctors led to a lack of focus on secondary fracture prevention. Efforts were needed to change the mindset of the orthopaedic doctors and subsequently improve awareness of the importance of secondary fracture prevention.

To ensure the sustainability of a FLS, our participants agreed on the need for a coordinator to help identify fragility fracture patients, refer them for bone health investigations, initiate treatment and provide follow-up care [22, 23]. However, In resource-limited countries, there is a need to implement the service with available resources, which is true to our situation. Therefore, job reallocation among the currently available manpower may be required to help with the implementation of the service.

Most doctors believed that nurses were the most suitable candidates to be FLS coordinators, consistent with previous findings [22]. However, our nurses lacked the confidence to take up this role as they felt that this role required a person with leadership qualities to effectively coordinate the team. At present, the majority of the registered nurses in Malaysia are diploma graduates and may perceive themselves as not being on par with other healthcare professional graduates. More training and supervision are needed to empower nurses to take on the role of FLS coordinator. Incentives may help encourage participation [11].

Pharmacists were included in this study due to their potential to become a FLS coordinator, as seen in other countries [24]. Pharmacists believed they were capable of assuming this role due to their expertise in medication knowledge. However, they were very short-staffed.

Particularly in our setting, doctors in primary care are unable to prescribe antiosteoporosis medications, posing a challenge for the transition of patients from secondary care to primary care [25]. Consequently, the majority of the fragility fracture patients were treated at the tertiary hospitals and their management was not decentralised. Moreover, the cost of the antiosteoporosis medications was often borne by these tertiary centres. This discrepancy was not reported in other FLS studies as there were no prescribing restrictions on antiosteoporosis medications for primary care doctors. Future studies may explore other general practitioners' views on secondary fracture prevention management and FLS, providing valuable insights to support policy changes facilitating treatment transition and continuation.

One of the limitations of our study was that it was performed in a tertiary teaching hospital. Given the diverse healthcare systems in Malaysia, the findings may not be generalised to institutions with different healthcare settings. Further work may be needed to explore barriers, especially in rural settings where bone mineral density machines may be lacking. We were only able to recruit one policymaker despite two rounds of emails. This may be because COVID-19 was still the main priority of the policymakers during the study period. Several pharmacists and nurses were not aware of FLS, and although the definition was provided during the interview, input from these participants was more limited due to the lack of understanding regarding this service. Our study only involved policymakers at the hospital level as they were the decision-makers for service implementation at the hospital. Nevertheless, future studies may explore stakeholders' views at a national level as it may help promote service implementation on a larger scale. Finally, our study mainly focused on the initial phases of FLS implementation, it would be beneficial to explore the barriers and facilitators of running FLS long-term in future studies.

## Conclusion

Our participants highlighted the necessity of a FLS to enhance secondary fracture prevention in Malaysia. However, they identified several challenges, including the lack of confidence among nurses in taking up the role of FLS coordinator, the need to implement the service within the constraints of limited resources, and the lack of prescribing rights for

antiosteoporosis medications among primary care physicians. Support from relevant stake-holders, improved FLS awareness, the presence of FLS coordinator and national policy were required to improve the sustainability of FLS. Providing training for nurses can help boost their confidence and encourage the uptake of the FLS coordinator role.

## Supporting information

**S1 Appendix. Full topic guide for different stakeholders.**
(PDF)

**S2 Appendix. Illustrative quotations of the study themes.**
(PDF)

## Author Contributions

**Conceptualization:** Min Hui Cheah, Pauline Siew Mei Lai, Terence Ong.

**Data curation:** Min Hui Cheah, Terence Ong.

**Formal analysis:** Min Hui Cheah, Pauline Siew Mei Lai, Terence Ong.

**Funding acquisition:** Terence Ong.

**Investigation:** Min Hui Cheah, Terence Ong.

**Methodology:** Min Hui Cheah, Pauline Siew Mei Lai, Terence Ong.

**Project administration:** Min Hui Cheah, Terence Ong.

**Resources:** Terence Ong.

**Software:** Pauline Siew Mei Lai, Terence Ong.

**Supervision:** Pauline Siew Mei Lai, Terence Ong.

**Validation:** Pauline Siew Mei Lai, Terence Ong.

**Visualization:** Pauline Siew Mei Lai, Terence Ong.

**Writing – original draft:** Min Hui Cheah.

**Writing – review & editing:** Min Hui Cheah, Pauline Siew Mei Lai, Terence Ong.

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
