## [Decision Letter · Decision Letter 0]

19 Mar 2024

PONE-D-24-04952Views of healthcare professionals regarding barriers and facilitators for a Fracture Liaison Service in MalaysiaPLOS ONE

Dear Dr. Ong,

Thank you for submitting your manuscript to PLOS ONE. After careful consideration, we feel that it has merit but does not fully meet PLOS ONE’s publication criteria as it currently stands. Therefore, we invite you to submit a revised version of the manuscript that addresses the points raised during the review process.

We look forward to receiving your revised manuscript.

Kind regards,

Osama Farouk

Academic Editor

PLOS ONE

Reviewers' comments:

Reviewer's Responses to Questions

**Comments to the Author**

1. Is the manuscript technically sound, and do the data support the conclusions?

Reviewer #1: No

Reviewer #2: Partly

2. Has the statistical analysis been performed appropriately and rigorously? 

Reviewer #1: Yes

Reviewer #2: No

3. Have the authors made all data underlying the findings in their manuscript fully available?

Reviewer #1: Yes

Reviewer #2: No

4. Is the manuscript presented in an intelligible fashion and written in standard English?

Reviewer #1: No

Reviewer #2: No

5. Review Comments to the Author

Reviewer #1: I would thank the editor for considering me as a reviewer for this submission in this esteemed journal.

I would thank the authors for their valuable effort in this research which discusses an important concern in improving FLS services.

The following points may be considered to improve the valuable study:

1. Although I believe that language should not be a barrier to knowledge transfer, I would ask the authors to improve the writing of the paper.

2. On page 2 line 48, it is better to mention the origin of this substantial service and discuss it more.

3. On page 2 line 50, the aim of this service is increasing the osteoporosis diagnosis, not the bone mineral densitometry itself.

4. On page 3 lines 60-63 were not relevant to the discussing topic.

5. The theory of planned behavior as the authors described it properly, predicts the intentions of an action/behavior. But in this study, as the topic and introduction indicate, the aim of the study was not the same and the asked questions were not in line with the claimed aim of the study. This will impress the method of the study widely.

6. I would prefer the transcript of the interview with recruited individuals to be prepared without any selection bias and as a supplementary file, not in the main draft of the paper.

7. I prefer to know the process of creating codes from transcripts and the way they were interpreted and analyzed in the data analysis part of the materials and methods.

8. The questions for the interview weren’t fully appropriate for the aim of the study.

Best Regards

Reviewer #2: In general, it’s a good idea but the major comment in this study is that this study was not written with referral to the focus group discussion with its details, there is a way of scientific writing of this qualitative study which is lacking in some points in his manuscript

The manuscript is needed to be written in some sections with referral to focus group discussions

Multiple linguistic corrections and revisions are needed

Abstract::

Line 30: Correct thirty to thirteen

Methods:

- The inclusion of policymakers in lines 107 to 109:

“In our setting, policymakers were the Director and the Deputy Directors of the hospital, and the relevant Head of Departments involved in the care of patients with fragility fractures” ….. this indicates more than 5 participants at least, while the actual included from the policymakers was one policymaker.

- Procedure section: It’s better to be replaced by focus group discussion setting.

-The details of the sitting were not mentioned as: who did the interview, who is the assistant, the place of interview, how many groups were made ‘…………….

- the lines from 153-156 are not clear

- The minimum number of focus group discussion is 4 to 8 or 6 to 12 to be ideal.

It’s better to delete the policymaker. He couldn’t be considered as a grup.

Results:

Table (1): the presentation of the age and length of working experience as mean and SD is not right

- The writing results in all themes as “some said”, “they have the opinion” or they said. All these expressions could be replaced by numbers and %. The opinion with the number in each group

- The titles of the tables are deficient.

6. PLOS authors have the option to publish the peer review history of their article (what does this mean?). If published, this will include your full peer review and any attached files.

Reviewer #1: **Yes: **Sepideh Hajivalizadeh

Reviewer #2: No

---

## [Author Response · Author response to Decision Letter 0]

30 Apr 2024

Dear Editor, 

Re: Re-submission of the manuscript: Views of healthcare professionals regarding barriers and facilitators for a Fracture Liaison Service in Malaysia

We, the authorship group, would like to thank the editor and the peer reviewers for their comments which were helpful and constructive. 

We have listed our responses to the comments raised. 

Reviewer #1: 

1. Although I believe that language should not be a barrier to knowledge transfer, I would ask the authors to improve the writing of the paper.

Response:

We have improved some of the sentences and checked that there are no grammatical errors. The changes made are shown as tracked changes in the ‘Revised manuscript with track changes” file.

2. On page 2 line 48, it is better to mention the origin of this substantial service and discuss it more.

Response:

We have included a few sentences to provide more context regarding the FLS.

“While fragility fractures are associated with significant morbidity and mortality, there remains a gap in the delivery of secondary fracture prevention [3, 4]. A Fracture Liaison Service (FLS) is a coordinator-based model developed to reduce subsequent fractures and mortality by increasing the diagnosis of osteoporosis and improving the treatment of osteoporosis [5, 6]. FLS was first introduced in Glasgow, Scotland in 1999 [6, 7]. Generally, FLS involves proactive identification of fragility fracture patients, bone health investigations, initiation of osteoporosis treatment where necessary, and follow-up to improve treatment adherence [6].” [Introduction, Page 2, Lines 50-55]

3. On page 2 line 50, the aim of this service is increasing the osteoporosis diagnosis, not the bone mineral densitometry itself.

Response:

The sentence was reworded to “A Fracture Liaison Service (FLS) is a coordinator-based model developed to reduce subsequent fractures and mortality by increasing the diagnosis of osteoporosis and improving the treatment of osteoporosis.” [Introduction, Page 2, Lines 50-52]

4. On page 3 lines 60-63 were not relevant to the discussing topic.

Response:

This paragraph summarises past FLS qualitative studies. Lines 60-63 [Introduction, Page 3, Lines 65-68] reported that the presence of FLS coordinators facilitated the implementation of FLS. We included this information as it aligned with our aim to explore the views of healthcare professionals regarding the barriers and facilitators for a FLS. No changes were made.

5. The theory of planned behavior as the authors described it properly, predicts the intentions of an action/behavior. But in this study, as the topic and introduction indicate, the aim of the study was not the same and the asked questions were not in line with the claimed aim of the study. This will impress the method of the study widely.

Response:

Our study aimed to explore the views of healthcare professionals regarding the barriers and facilitators for a FLS in Malaysia, especially to facilitate its implementation in this tertiary hospital. We used the theory of planned behaviour as the conceptual framework for this study as the participants’ intention to participate in the FLS and willingness (behaviour) to implement a FLS would be affected by their attitudes, subjective norms and perceived behavioural control. These are factors that either encourage or hinder them from implementing FLS. [Methods, Page 4, Lines 86-90]

Added “Based on the TPB determinants, the relevant stakeholders’ attitudes about the FLS model were explored by exploring their opinions on the current delivery of secondary fracture prevention and the importance of secondary fracture prevention via a FLS. Subjective norm is an individual’s perception of whether their significant others (eg. family and friends) approve or disapprove of the behaviour. Participants’ perceived behavioural control was identified by exploring their barriers and facilitators to implementing FLS, which would contribute to the sustainability of the FLS. Intentions would be formed from these aspects thus affecting their willingness to set up the FLS.” [Methods, Page 4, Lines 90-96]

6. I would prefer the transcript of the interview with recruited individuals to be prepared without any selection bias and as a supplementary file, not in the main draft of the paper.

Response:

We inserted the participants’ quotes in our manuscript for the following reasons: 1) they can serve as concrete examples that illustrate the themes, 2) the quotes support our interpretations and help to validate the conclusions drawn from the data analysis, 3) they allow a deeper understanding of the participants’ perspectives, experiences, and emotions, 4) they provide context and nuance to the data. 

We are aware of the possible selection bias and therefore had regular meetings (twice a month) to discuss the analysis until a consensus was reached. The original interview transcripts could not be provided as there was some potentially identifiable information, however, we have prepared a draft of our analysis with the original quotes as a supplementary file (S2 Appendix).

7. I prefer to know the process of creating codes from transcripts and the way they were interpreted and analyzed in the data analysis part of the materials and methods.

Response:

We have rewritten the process of data analysis.

“Participants’ demographic details were summarised to maintain anonymity. Qualitative data were managed using Atlas.ti software version 22 (Scientific Software Development GmbH, Berlin, Germany). All audio recordings were transcribed by a transcriptionist, and cross-checked by MHC for accuracy and completeness. Interviewee transcript review was not performed. Researchers involved in the analysis were a geriatrician (TO), an academic pharmacist (PSML) and a postgraduate researcher (MHC). All researchers were conscious of their personal views and biases regarding FLS implementation. The team had regular meetings to reflect on and discuss the analysis. 

MHC independently reviewed and coded the transcripts. A meeting was carried out after the first three transcripts were coded. PSML and TO reviewed the quotes and codes identified to ensure the reliability and consistency of the codes. The emergence of patterns from open coding was observed. MHC then continued to analyse the remaining transcripts. The research team met fortnightly to review the codes. The codes were merged into categories (subthemes) and a network was built to identify the relationships between codes, categories, and concepts derived from open coding. The subthemes were mapped based on the TPB as we identified a close relationship between the themes that emerged and the TPB. A draft codebook was generated and discussed among the researchers, and the subthemes and themes were further refined based on the discussions until the codebook was finalised. Disagreements were resolved through consensus.”[Methods, Page 7, Lines 161-176]

8. The questions for the interview weren’t fully appropriate for the aim of the study.

Response:

The topic guide was developed based on the TPB. Questions 1-4 in the topic guide explored participants’ attitudes regarding the current secondary fracture prevention and their understanding of the FLS. Their views of the current delivery may affect their attitude regarding the need and importance of a FLS thus facilitating or impeding the FLS implementation. Questions 5-9 explored the barriers and facilitators of FLS implementation. As FLS is a coordinator-based model, we included a few questions to understand the participants’ views on the coordinator. As mentioned in the methods section [Methods, Page 5, Lines 125-126], the same topic guide was modified to cater for the different stakeholders. We have prepared the original topic guide as a supplementary file for the reviewers’ and readers’ reference (S1 Appendix).

Reviewer #2: 

1. In general, it’s a good idea but the major comment in this study is that this study was not written with referral to the focus group discussion with its details, there is a way of scientific writing of this qualitative study which is lacking in some points in his manuscript

The manuscript is needed to be written in some sections with referral to focus group discussions

Response:

This qualitative study only utilised semi-structured one-on-one in-depth interviews [Methods, Page 6, Line 152-153]. We did not carry out any focus group discussions. we have included more details on how the interviews were carried out [Methods, Page 6-7, Line 153-159].

We have made some minor modifications to the manuscript as recommended in the COREQ checklist. Please find attached the checklist for your reference.

2. Multiple linguistic corrections and revisions are needed

Response:

We have improved some of the sentences and checked that there are no grammatical errors. The changes made are shown as tracked changes in the ‘Revised manuscript with track changes” file.

3. Abstract: 

Line 30: Correct thirty to thirteen

Response:

Line 30 “Thirty participants [doctors (n=13), nurses (n=8), pharmacists (n=8), and policymakers (n=1)] with 2-28 years of working experience were recruited.”

The total number of participants was thirty. Thirteen is the number of doctors that participated in this study. No changes were made.

4. Methods:

- The inclusion of policymakers in lines 107 to 109:

“In our setting, policymakers were the Director and the Deputy Directors of the hospital, and the relevant Head of Departments involved in the care of patients with fragility fractures” ….. this indicates more than 5 participants at least, while the actual included from the policymakers was one policymaker.

Response:

The sentence above explains the targeted policymakers in our setting. We mentioned in our results that only 1 out of 11 policymakers approached agreed to participate in the interview [Results, Page 8, Line 185-186]. In our limitations, we stated that “We were only able to recruit one policymaker despite sending out two rounds of emails inviting them to participate. This may be because COVID-19 was still the main priority of the policymakers during the study period.” [Discussions, Page 17, Line 436-437] No changes were made.

5. Procedure section: It’s better to be replaced by focus group discussion setting.

Response:

This has been addressed in point 1. We carried out individual interviews instead of focus group discussions as it allowed us to delve deeply into individual perspectives, experiences, and opinions without the influence or distraction of group dynamics. Besides, most of our participants had busy schedules that made it difficult to coordinate group meetings.

6. The details of the sitting were not mentioned as: who did the interview, who is the assistant, the place of interview, how many groups were made ‘…………….

Response:

Thanks for the suggestions, we have added information regarding the interview details. [Methods, Page 6-7, Line 152-157]

“Semi-structured in-depth interviews were conducted virtually as this study was conducted during the COVID-19 pandemic. The interviews were conducted one-on-one with the participants by MHC, a postgraduate researcher who has undergone training in doing qualitative research. No relationship was established between the interviewer and the participants before the study commencement. The purpose of the interview was explained to the participants at the beginning of the interview. Interviews lasted 30 to 60 minutes. Field notes on non-verbal cues and interview dynamics were recorded.”

7. the lines from 153-156 are not clear

Response:

The data analysis was expanded in detail. [Methods, Page 7, Line 161-176]

“Participants’ demographic details were summarised to maintain anonymity. Qualitative data were managed using Atlas.ti software version 22 (Scientific Software Development GmbH, Berlin, Germany). All audio recordings were transcribed by a transcriptionist, and cross-checked by MHC for accuracy and completeness. Interviewee transcript review was not performed. Researchers involved in the analysis were a geriatrician (TO), an academic pharmacist (PSML) and a postgraduate researcher (MHC). All researchers were conscious of their personal views and biases regarding FLS implementation. The team had regular meetings to reflect on and discuss the analysis. 

MHC independently reviewed and coded the transcripts. A meeting was carried out after the first three transcripts were coded. PSML and TO reviewed the quotes and codes identified to ensure the reliability and consistency of the codes. The emergence of patterns from open coding was observed. MHC then continued to analyse the remaining transcripts. The research team met fortnightly to review the codes. The codes were merged into categories (subthemes) and a network was built to identify the relationships between codes, categories, and concepts derived from open coding. The subthemes were mapped based on the TPB as we identified a close relationship between the themes that emerged and the TPB. A draft codebook was generated and discussed among the researchers, and the subthemes and themes were further refined based on the discussions until the codebook was finalised. Disagreements were resolved through consensus”

8. The minimum number of focus group discussion is 4 to 8 or 6 to 12 to be ideal.

It’s better to delete the policymaker. He couldn’t be considered as a group.

Response:

This has been explained in points 1, 4 and 5. Although only one policymaker could be recruited, thankfully the policymaker who participated in the interview was an important individual in our hospital who oversees all clinical service delivery. He provided valuable insights on factors that might influence the decision-making process regarding a new service implementation. Therefore, we would like to keep him in our study.

9. Results:

Table (1): the presentation of the age and length of working experience as mean and SD is not right

Response:

We have changed the presentation of age and length of working experience to range instead. [Results, Page 8, Table 2]

10. The writing results in all themes as “some said”, “they have the opinion” or they said. All these expressions could be replaced by numbers and %. The opinion with the number in each group.

Response:

Usually, in qualitative studies, numbers and % are not reported as our main priority is to understand the depth and nuances of participants' perspectives without assigning weightage to individual responses. No changes were made.

11. The titles of the tables are deficient.

Response:

“Table 1. Topic guide used” changed to “Table 1. Topic guide for semi-structured in-depth interviews” [Methods, Page 5, Table 1]

“Table 2. Baseline demographic characteristics of participants” changed to “Table 2. Baseline demographic characteristics of study participants” [Results, Page 8, Table 2]

“Table 3. Themes that emerged.” changed to “Table 3. Themes emerged from qualitative data analysis” [Results, Page 9, Table 3]

---

## [Decision Letter · Decision Letter 1]

14 Jun 2024

PONE-D-24-04952R1Views of healthcare professionals regarding barriers and facilitators for a Fracture Liaison Service in MalaysiaPLOS ONE

Dear Dr. Ong,

Thank you for submitting your manuscript to PLOS ONE. After careful consideration, we feel that it has merit but does not fully meet PLOS ONE’s publication criteria as it currently stands. Therefore, we invite you to submit a revised version of the manuscript that addresses the points raised during the review process.

We look forward to receiving your revised manuscript.

Kind regards,

Osama Farouk

Academic Editor

PLOS ONE

Journal Requirements:

Reviewers' comments:

Reviewer's Responses to Questions

**Comments to the Author**

1. If the authors have adequately addressed your comments raised in a previous round of review and you feel that this manuscript is now acceptable for publication, you may indicate that here to bypass the “Comments to the Author” section, enter your conflict of interest statement in the “Confidential to Editor” section, and submit your "Accept" recommendation.

Reviewer #1: (No Response)

Reviewer #2: All comments have been addressed

2. Is the manuscript technically sound, and do the data support the conclusions?

Reviewer #1: Yes

Reviewer #2: Yes

3. Has the statistical analysis been performed appropriately and rigorously? 

Reviewer #1: Yes

Reviewer #2: Yes

4. Have the authors made all data underlying the findings in their manuscript fully available?

Reviewer #1: Yes

Reviewer #2: (No Response)

5. Is the manuscript presented in an intelligible fashion and written in standard English?

Reviewer #1: Yes

Reviewer #2: Yes

6. Review Comments to the Author

Reviewer #1: Dear Editor,

I would like to thank you for considering me as the reviewer of this manuscript.

Dear Authors,

In the “method” section, lines 95 to 101, I suggest first explaining each construct of the theory properly and then determining each equivalent in your study. You explained the equivalents of two of them and the definition of one. Explain all in both ways completely so the reader can understand the whole concept and assign each part to the assessed equivalents in your study.

Best Regards.

Reviewer #2: Dear authors

Thank you for the done revisions. After reevaluation of the revised version, no more revisions are needed.

Best regards

7. PLOS authors have the option to publish the peer review history of their article (what does this mean?). If published, this will include your full peer review and any attached files.

Reviewer #1: **Yes: **Sepideh Hajivalizadeh

Reviewer #2: **Yes: **Dalia G Mahran

---

## [Author Response · Author response to Decision Letter 1]

18 Jun 2024

Dear Editor, 

Re: Re-submission of the manuscript: Views of healthcare professionals regarding barriers and facilitators for a Fracture Liaison Service in Malaysia

We, the authorship group, would like to thank the editor and the peer reviewers for their comments which were helpful and constructive. 

We have listed our responses to the comments raised. 

Reviewer #1: 

1. In the “method” section, lines 95 to 101, I suggest first explaining each construct of the theory properly and then determining each equivalent in your study. You explained the equivalents of two of them and the definition of one. Explain all in both ways completely so the reader can understand the whole concept and assign each part to the assessed equivalents in your study.

Response:

We have modified the paragraph accordingly. [Methods, Page 4, Lines 90-99]

“Attitude refers to an individual’s positive or negative evaluation of performing a behaviour. In our study, we assessed stakeholders' attitudes by exploring their opinions on the current delivery of secondary fracture prevention and the perceived importance of implementing an FLS. Subjective norm involves the perceived social pressure to perform or not perform a behaviour. We investigated this construct by understanding stakeholders' perceptions of whether significant others (such as family, friends, and colleagues) support or oppose the implementation of the FLS model. Perceived behavioural control is the perceived ease or difficulty of performing a behaviour, reflecting past experiences and anticipated obstacles. We identified this by examining the barriers and facilitators stakeholders perceive in implementing and sustaining the FLS model. These constructs together shape stakeholders' intentions, indicating their willingness and readiness to implement and sustain the FLS model.”

---

## [Editor Report · Decision Letter 2]

15 Jul 2024

Views of healthcare professionals regarding barriers and facilitators for a Fracture Liaison Service in Malaysia

PONE-D-24-04952R2

Dear Dr. Ong,

We’re pleased to inform you that your manuscript has been judged scientifically suitable for publication and will be formally accepted for publication once it meets all outstanding technical requirements.

Kind regards,

Osama Farouk

Academic Editor

PLOS ONE

Additional Editor Comments (optional):

The authors respond to the reviewer's comments in satisfactory manner.
---

## [Editor Report · Acceptance letter]

18 Jul 2024

PONE-D-24-04952R2 

PLOS ONE

Dear Dr. Ong, 

I'm pleased to inform you that your manuscript has been deemed suitable for publication in PLOS ONE. Congratulations! Your manuscript is now being handed over to our production team.

Kind regards, 

on behalf of

Dr. Osama Farouk 

Academic Editor

PLOS ONE